# Active Carbon-Based Electrode Materials from Petroleum Waste for Supercapacitors

**Abdualilah Albaiz, Muhammad Alsaidan, Abdullah Alzahrani, Hassan Almoalim, Ali Rinaldi \*,†
and Almaz S. Jalilov \***

Department of Chemistry and Interdisciplinary Research Center for Advanced Materials, King Fahd University of Petroleum and Minerals, Dhahran 31261, Saudi Arabia

\* Correspondence: ali.rinaldi@tum.de (A.R.); jalilov@kfupm.edu.sa (A.S.J.)

† Current Address: Faculty of Chemistry, School of Natural Sciences, Technical University of Munich, Lichtenbergstrasse 4, 85748 Garching bei München, Germany.

**Abstract:** A supercapacitor is an energy-storage device able to store and release energy at fast rates with an extended cycle life; thus, it is used in various electrical appliances. Carbon materials prepared above 800 °C of activation temperatures are generally employed as an electrode material for supercapacitors. Herein, we report carbon materials prepared from a low-cost petroleum waste carbon precursor that was activated using KOH, MgO, and Ca(OH)$_2$ only at 400 °C. Electrode materials using low-temperature activated carbons were prepared with commercial ink as a binder. The cyclic voltammetry and galvanostatic charge–discharge were employed for the electrochemical performance of the electrodes, and studied in a 3-electrode system in 1 M solutions of potassium nitrate (KNO$_3$) as electrolyte; in addition, the supercapacitive performance was identified in a potential window range of 0.0–1.0 V. The best-performance activated carbon derived from vacuum residue with a specific surface area of 1250.6 m$^2$/g exhibited a specific capacitance of 91.91 F/g.

**Keywords:** activated carbon; vacuum residue; supercapacitor; carbon nanotechnology

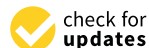



## 1. Introduction

Supercapacitors, i.e., electrochemical capacitors, have a high-power density, fast charging rate, cyclability, and low cost of operation [1,2]. Thus, they have found many uses in electronics, energy management, mobile electrical systems, as well as industrial power arrays [3,4]. In an electrochemical double-layer supercapacitor, the energy is stored in the electrostatic charge separation on the surface between the electrode and electrolyte. High porosity and good mechanical properties allow porous carbon materials to be widely used as electrode materials [5].

In a supercapacitor, energy is stored either by an electrochemical double layer in the electrode–electrolyte interphase (nonfaradaic) and/or by reduction–oxidation reactions (faradaic) on electrode surfaces [6–12]. For electrochemical double-layer capacitors (EDLC), ions in the electrolyte will accumulate at the surface of the solid electrode materials. The capacitance (CDL) of the electrode depends on the layer thickness where the ions and solvent molecules reside, d; and the surface area of the electrode, A (see equation below). ε and εr are the permittivity of the vacuum and the solvent, respectively.

$$CDL = \varepsilon \times \varepsilon r \, A/d$$

For high-energy and power-density EDLC, it is imperative for the electrode to possess a high surface area for cations and anions to accumulate. Porous carbons, such as graphene-based nanocarbons, carbon nanotubes, and activated carbons, are used as electrodes in EDLC supercapacitor devices due to their high surface area, high electrical conductivity, and electrochemical stability. The low-cost and established electrode fabrication technologies in the industry create

a sustainable approach for using carbons as electrodes in supercapacitor devices. Activated carbon with a BET surface area of 3150 $m^2$/g was reported to show a capacitance of 312 F/g. This capacitance translates into a specific capacitance value of 9.9 $\mu F/cm^2$ [13].

Porous carbons from various synthesis routes have been explored as high-surface area electrodes for supercapacitors [14,15]. Asphalt, a petroleum waste, has been used as a precursor to high-surface area carbons that were used as electrodes in batteries and supercapacitors [16–18]. Asphalt is a low-cost material obtained from the heaviest fraction of crude oil. Asphalt contains some amount of volatile organic species that are removed during a carbonization process at about >400 °C. Asphalt-derived carbons with surface areas of >4000 $m^2$/g were achieved after a sequential process of removal of volatile organic followed by high-temperature activation with KOH [19,20].

This energy-intensive activation step to create a high surface area can be compensated by using alternative and inexpensive activating agents from industrial waste. Coal-fired power plants, on the other hand, are the large-point sources of carbon dioxide ($CO_2$) [21] and generate solid waste such as fly ash and bottom ash, which mainly contain lime (CaO) and magnesium oxide (MgO) [22,23]. However, using such alternative and inexpensive mineral feedstock from industrial waste as activating agents is challenging due to the slow chemical kinetics and high activation energy [10]. Furthermore, most of the methods of activation of carbons require high temperatures, usually above 800 °C. The combination of using asphalts as petroleum waste as a carbon source and solid industrial waste as activating agents can be an attractive cost-effective route towards synthesizing high-surface-area carbons for supercapacitor application.

In this work, we aim to use vacuum residue (VR) as the carbon precursor for generating low-density porous carbon materials with KOH, MgO, and Ca(OH)$_2$ activation under mild conditions of activation at 400 °C. The resultant activated carbon materials will be tested as electrode materials for supercapacitors.

## 2. Materials and Methods

Materials. Vacuum residue (VR) was acquired from the local Ras Tanura refinery in the Eastern Province of Saudi Arabia. The average composition of VR is >85 wt% carbon, ~10 wt% hydrogen, ~4 wt% sulfur, and <1 wt% nitrogen; Vulcan black, calcium hydroxide (Ca(OH)$_2$), magnesium oxide (MgO), and potassium hydroxide (KOH) were purchased from Millipore-Sigma and used without further purification; ultrapure water processed from the Milli-Q (Milford, MA, USA) system.

Synthesis. The preparation of porous carbon materials from VR was conducted in one step using activating agents, such as calcium hydroxide (Ca(OH)$_2$), magnesium oxide (MgO), and potassium hydroxide (KOH). Briefly, different ratios of VR and the activation agents are mixed and placed in a horizontal tubular furnace and heat treated under an inert atmosphere at 400 °C for 4 h. The activation procedures are employed with the ratio = 1:4 (between VR: activating agent), denoted as VR-KOH, VR-Ca(OH)$_2$, VR-MgO. After the activation, the samples are washed with deionized water until a neutral pH and oven-dried at 110 °C overnight.

Characterization. X-ray photoelectron spectroscopic (XPS) analyses were obtained on a Thermo Scientific Escalab 250Xi spectrometer with Al Ka (1486.6 eV) as the x-ray source and an operating resolution of 0.5 eV. X-rays with a 650 μm beam and pass energy of 100 eV are used for the survey scan, and 30 eV is used for the high-resolution scans. High-resolution spectra for binding energies spectra were centered at 284.8 eV, corresponding to the C 1s of the graphitic carbon (C–C/C=C). SDT Q600 (TA instruments) was used for the thermal gravimetric analysis (TGA). Typically, the measurement was performed by heating ~10 mg weight of samples up to 900 °C in aluminum pans at a constant heating rate of 10 °C min$^{-1}$ under N$_2$ flow (99.999% purity). A Quattro ESEM 400 high-resolution field emission scanning electron microscope (SEM) at 20 keV is used for the SEM images and energy dispersive x-ray analysis (EDX). Brunauer–Emmett–Teller (BET) analysis was performed to investigate the texture of the carbon materials using the Quantachrome Autosorb-

3b. Prior to the $N_2$ physisorption measurements, the samples were activated at 300 °C for 24 h under a vacuum.

Electrode Preparation. Graphite foil was cut in circular form with a diameter of 1 cm and used as the working electrode. Typically, 1 gr of ink is mixed with the ~30 mg of active sample and the suspension is sonicated for 10 min. In all, ~20 mg of well-dispersed suspension is drop-casted on the graphite disc and dried at 85 °C for 48 h.

## 3. Results and Discussion

To examine the efficiency of the activating agents and the composition of the final materials, TGA was performed under air for all the materials after the activation step. It is expected that under an air atmosphere, the carbon content will be combusted leaving only the metal oxides. VR-KOH exhibited a major weight-loss profile at 350–470 °C, which corresponds to 93% weight of its original weight. This step is due to the combustion of carbon as shown in Figure 1b. On the other hand, both VR-Ca(OH)$_2$ and VR-MgO exhibit distinct thermal behavior as shown in Figure 1c,d, respectively. The major weight loss step for VR-Ca(OH)$_2$ occurred at 600–710 °C, corresponding to the combustion of carbon components in 32% of total weight. The remaining 68% is due to the CaO content. The thermal weight loss profile for VR-MgO reveals the major weight loss step at 350–500 °C, due to the combustion of carbon. This constitutes 17% of the total weight, pointing to 83% of the MgO composition in VR-MgO (Figure 1d).

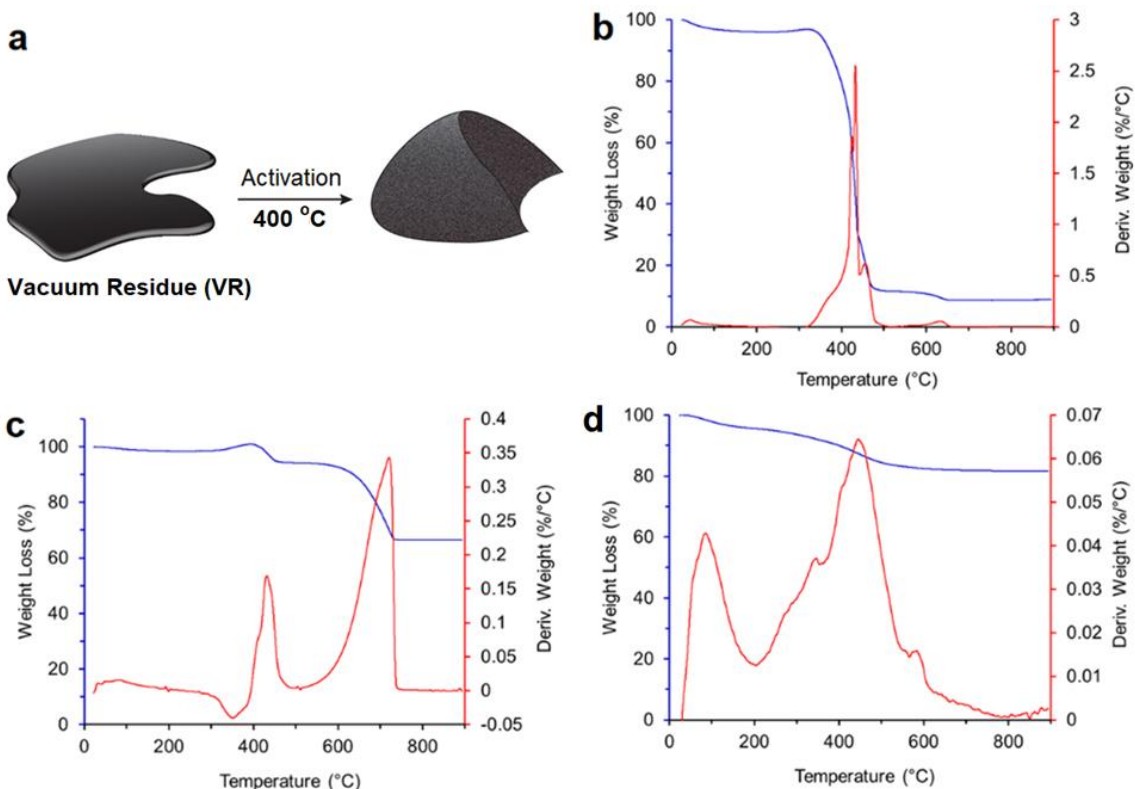

**Figure 1.** (**a**) Schematic representation of petroleum waste activation for porous activated carbon materials. Thermogravimetric curves of (**b**) VR-KOH, (**c**) VR-Ca(OH)$_2$, and (**d**) VR-MgO (the heating rate at 10 °C min$^{-1}$, under air).

The morphology of the activated carbons was investigated using SEM using either SE or BSE signals. In general, the morphology of the activated carbons herein is different than carbon black, where carbon nanoparticles form an aggregate. Carbon samples activated with either KOH, Ca(OH)$_2$, or MgO exhibit irregular shapes of micron-sized particles (Figure 2).

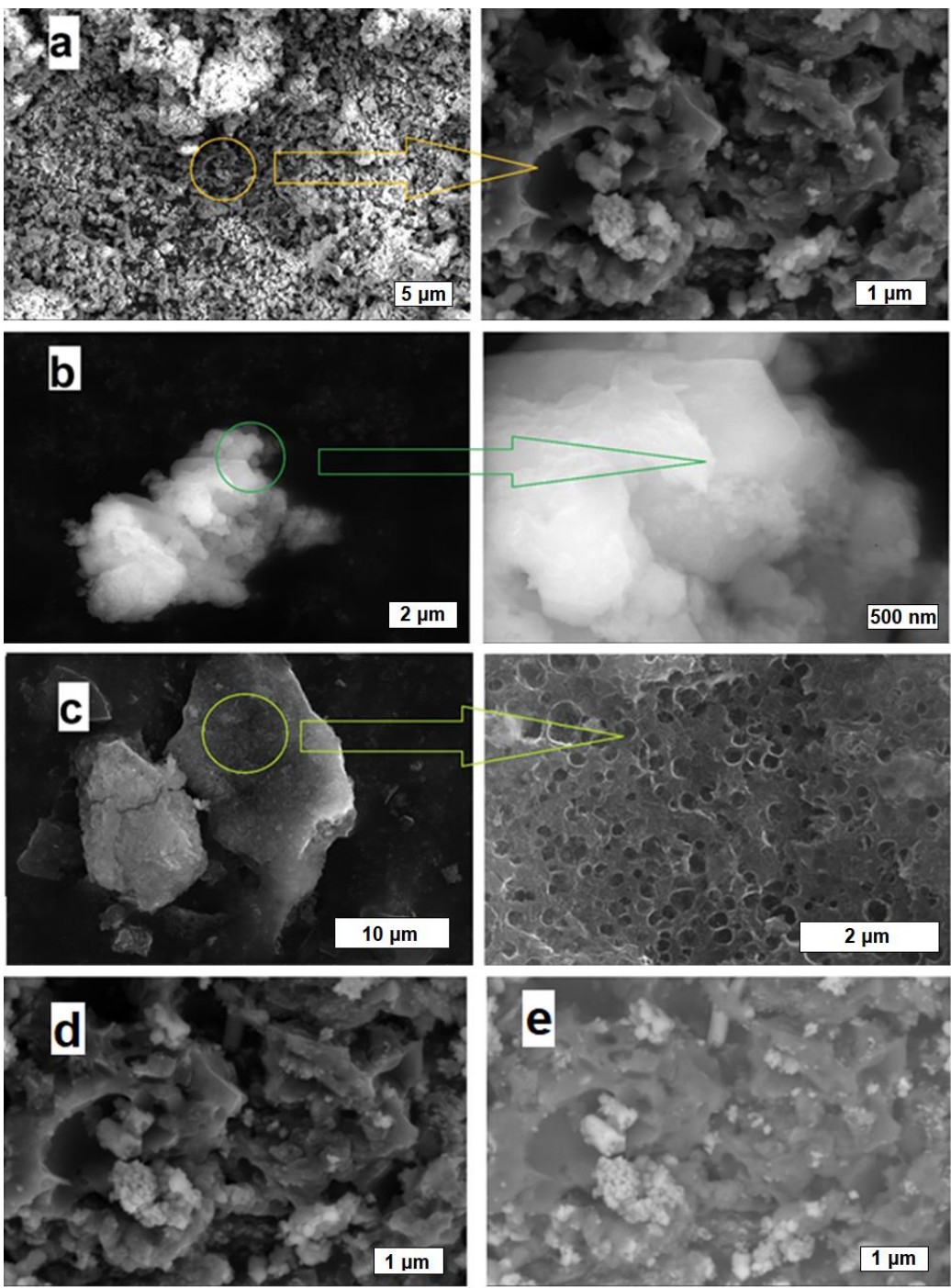

**Figure 2.** SEM images of (**a**) VR-KOH, (**b**) VR-Ca(OH)$_2$, (**c**) VR-MgO; and (**d**) SE and (**e**) BSE imaging mode of VR-KOH.

SEM analysis for VR-KOH (Figure 2a) shows a broad range of particle size of about 0.5 to 5 um. Irregular-shaped cavities were observed on the particles with sizes of about 50 to 500 nm. Inorganic materials were also observed in the sample dispersed on the carbon particle surfaces and cavities as individual particles or agglomerates. These inorganic materials on the carbon sample can be seen from their brighter contrast in the BSE and SE imaging modes (additional SEM images). It is safe to say that the inorganic particles are mainly composed of KOH, the activating agent added to the asphalt precursor. This agrees with the TGA measurement, which revealed a high carbon content in VR-KOH. Due to the low carbon content of the VR-Ca(OH)$_2$ sample and the non-conducting nature of Ca(OH)$_2$ particles, SEM imaging analyses were performed with a low-vacuum mode under water vapor. Nevertheless, VR-Ca(OH)$_2$ shows agglomerated nanoparticles of mainly CaO, with sizes in the range of 50–500 nm (Figure 2b). The morphology of the VR-MgO sample shows a greater amount of macropores compared to the other two activated samples as shown in Figure 2c. The macropores present in VR-MgO are in the range of 100 nm to 5 um. The porosity produced on the activated sample depends on the activating agent. This indicates the different activating or oxidizing power of the activating agents. In addition, other factors such as the initial dispersion of the activating agents on asphalt may also lead to the varying progression of pore formation. The EDX measurements at 20 kV addressed the EDX-average value of the samples' sulfur content. The measurements on all of the activated samples show a low content of sulfur. The VR-MgO sample showed the highest sulfur content around 1.0 wt%, while both the VR-KOH and VR-Ca(OH)$_2$ samples show less than 0.1 wt%. EDX measurements were performed with a minimum of five spots for every sample.

A more macroscopic measurement of the activated samples' texture was obtained from N$_2$ adsorption isotherms at 77 K (Figure 3). The porosity is generated as a result of asphalt (vacuum residue) oxidation by the chemical activating agents under an inert atmosphere. A carbon black sample (Vulcan black) adsorption isotherm was used as a benchmark to compare adsorption isotherms of the VR-derived carbons. The carbon black sample has no microporosity and has only mesopores due to spaces between the carbon black nanoparticles. The BET surface areas were calculated from the adsorption isotherms as a measure of the total surface areas of the samples. The BET total surface area is the sum of the external surface area and the micropore surface area. The t-plot method was used to determine the external surface area (Table 1). In the t-plot method, the statistical thickness of the adsorbed N$_2$ onto the carbon surface is calculated as a function of the adsorbed volume. More adsorption at a higher P/Po will give higher thickness. The deviation will from linearity corresponds to micropore filling. The micropore area was then calculated by subtracting the BET total area from the external area. The total pore volume was calculated from the amount of N$_2$ adsorbed at a relative pressure of P/Po = 0.99. This pore volume corresponds to micro and mesopores filled until the N$_2$ gas reaches full condensation on the sample. The carbon black reference VR-Ca(OH)$_2$ and VR-MgO samples exhibit a similar adsorption isotherm feature with insignificant N$_2$ adsorption at P/Po < 0.01. This indicates a negligible amount of micropores. On the contrary, the VR-KOH sample shows appreciable adsorption at P/Po < 0.01, indicating abundant microporosity. Further analysis of the isotherms shows that the VR-KOH sample has a high BET total surface area and high micropore area. The external surface areas of the activated samples are of the same order of magnitude with some variation. However, the VR-MgO sample has about 40% more external area than the other two activated samples. This is supported by the more abundant large pores in the VR-MgO sample as seen in the SEM images (Figure 2c). KOH is better than either LiOH and NaOH in producing microporosity at 400 °C than either Ca(OH)$_2$ or MgO.

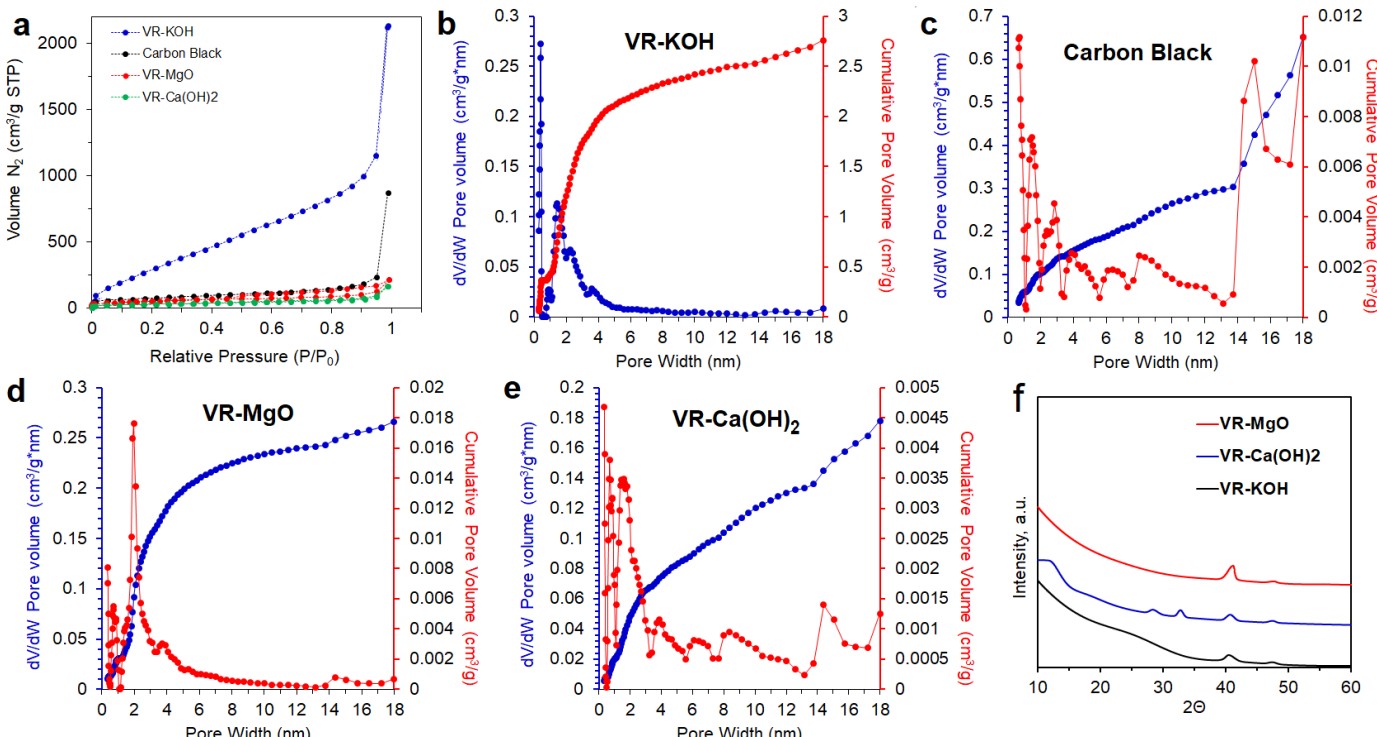

**Figure 3.** (**a**) Comparative $N_2$-adsorption isotherms classified as type I for VR-KOH, carbon black, VR-MgO, and VR-Ca(OH)$_2$; pore-size distribution curves for (**b**) VR-KOH, (**c**) carbon black, (**d**) VR-MgO, and (**e**) VR-Ca(OH)$_2$ determined using the NLDFT and (**f**) XRD pattern of VR-KOH, VR-MgO, and VR-Ca(OH)$_2$.

Textural parameters and specific surface areas for all the samples are summarized in Table 1. The $N_2$-adsorption isotherms are also shown in Figure 3 concerning a commercial carbon black. Micro- and mesoporous structures of the samples were also evident from the pore size distributions obtained by applying the NLDFT depicted in Figure 3b–e. VR-KOH and VR-MgO possess pore sizes in the range of 1–8 nm. On the other hand, a much wider pore size distribution range of up to 18 nm was observed for both VR-Ca(OH)$_2$ and carbon black. The XRD pattern shown in Figure 3f, reveals the relatively intense broad peak at the lower angle region of ~25 for VR-KOH, indicating a higher degree of graphitization in comparison to both VR-MgO and VR-Ca(OH)$_2$. All the samples possess the peak at ~40 assigned for (10) band reflections coming from the polyaromatic structure of asphaltenes, indicating a not complete carbonization of the VR at 400 °C activations. VR-Ca(OH)$_2$ also shows peaks at ~28 and ~33, demonstrating the presence of crystalline CaO.

**Table 1.** $N_2$-physisorption analysis data.

| Sample | $S_{BET}$ [1,5] [m$^2$ g$^{-1}$] | $V_{total}$ [2] [cm$^3$ g$^{-1}$] | Micropore Area [3] [m$^2$ g$^{-1}$] | External Surface Area [4] [m$^2$ g$^{-1}$] |
|---|---|---|---|---|
| VR-KOH | 1250.6 | 5.3 | 1146 | 103.9 |
| VR-Ca(OH)$_2$ | 99.8 | 0.259 | 5.33 | 94.47 |
| VR-MgO | 179.8 | 0.193 | 36.3 | 143.5 |
| Carbon black | 266.9 | 0.363 | 50.2 | 216 |

[1] BET area can be considered as the total surface area. [2] Taken at P/Po = 0.99. [3] Pore with diameter < 2 nm, when there is microporosity; BET area = micropore area + external surface area (t-plot). [4] Pore with diameter > 2 nm. [5] Repeated measurements of representative sample show about 14% standard deviation of BET values.

XPS analysis of the chemical surface composition of materials is summarized in Figure 4 and Table 2. Surface characterization of VR-KOH reveals 84.23% of carbon and 11.85% oxygen content with the amount of sulfur present at 2.09% and nitrogen at 1.83%. VR-Ca(OH)$_2$ possesses 42.75% of carbon, 39.09% of oxygen, and 16.85% calcium, with the

amount of nitrogen at 1.32%. VR-MgO possesses 34.09% of carbon, 52.69% of oxygen, and 13.22% magnesium. High-resolution C 1s XPS spectra with fitted deconvolution of the peaks for VR-KOH show the majority of carbon is in aromatic and aliphatic carbon form, whereas both VR-Ca(OH)$_2$ and VR-MgO, in addition to aromatic carbons, also possess carbonate forms of carbon; see Figure 4. It is noteworthy to mention that the main peak for all the samples at ~284.8 eV corresponds to graphitic carbon groups. The high content of oxygen, the appearance of carbonates, and the absence of aliphatic carbons in VR-Ca(OH)$_2$ and VR-MgO suggest extensive oxidation of the asphalt carbons by the Ca(OH)$_2$ and MgO as opposed to KOH. Due to their high thermal stability, the Ca- and Mg- carbonates are still present even after the activation step at 400 °C. The details of the fitting results concerning the corresponding carbon functional groups are shown in Figure 4.

**Table 2.** Elemental composition of VR-KOH, VR-Ca(OH)$_2$, and VR-MgO estimated from XPS data.

| Sample | C% | O% | N% | S% | Ca% | Mg% |
|---|---|---|---|---|---|---|
| VR-KOH | 84.23 | 11.85 | 1.83 | 2.09 | - | - |
| VR-Ca(OH)$_2$ | 42.75 | 39.09 | 1.32 | - | 16.85 | - |
| VR-MgO | 34.09 | 52.69 | - | - | - | 13.22 |

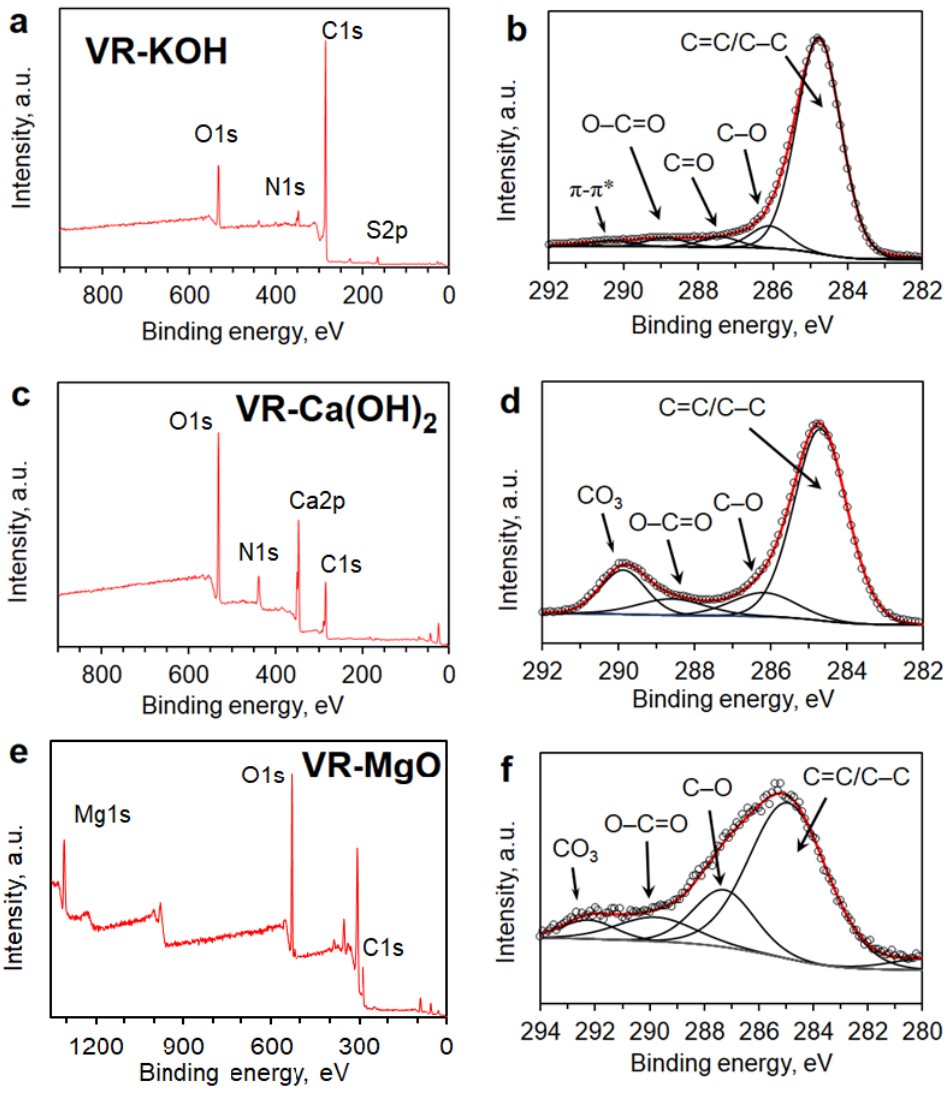

**Figure 4.** XPS survey spectra for (**a**) VR-KOH, (**c**) VR-Ca(OH)$_2$, and (**e**) VR-MgO. XPS high-resolution C 1s spectra for (**b**) VR-KOH, (**d**) VR-Ca(OH)$_2$, and (**f**) VR-MgO.

XPS, TGA, and $N_2$-adsorption isotherm analysis point out that the extent of the carbonaceous oxidation and the resulting microporosity depends on activating agents. KOH is the most efficient in generating micropores, while $Ca(OH)_2$ seems to extensively oxidize the carbonaceous materials in asphalt without micropores. It is also plausible to consider that the micropores were generated at an early stage in $Ca(OH)_2$ and MgO-activated samples. As such, with progressive oxidation in the activation step, the micropores enlarge and merge into meso- and macropores. Further study is required to elucidate the pore-formation steps and to find optimum conditions using the chemical agents or their combination. In addition, owing to the higher microporosity, VR-KOH was further analyzed using Raman spectroscopy as an indication of the degree of graphitization after the low-temperature activation step. The Raman spectra shown in Figure 5 compares VR-KOH as the sample with the highest carbon yield and the most porous with graphite powder. As can be seen in the figure, the VR-KOH sample exhibits a broad and pronounced D-band centered at ~1350 cm$^{-1}$. Furthermore, the intensity ratio of the D-band with the graphitic signature band (G-band) at ~1580 cm$^{-1}$ is much larger for VR-KOH than the graphite powder sample, which indicates a defective carbon material and a significant amount of amorphous carbon present in VR-KOH.

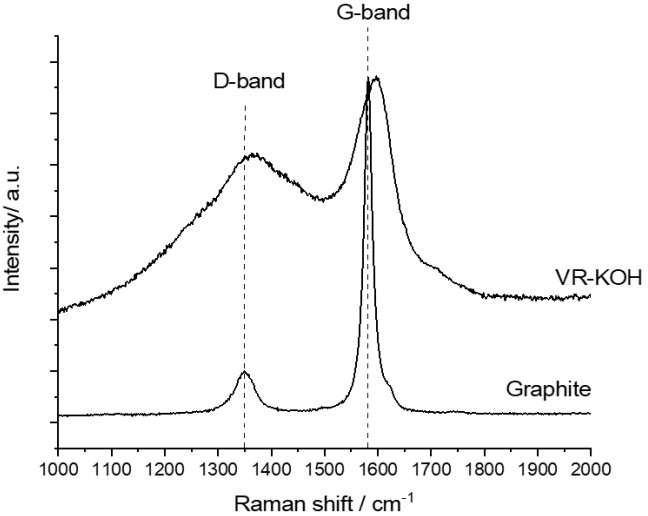

**Figure 5.** Raman spectra for VR-KOH and graphite powder.

Ongoing research in EDLC focuses mainly on increasing the capacitance by tuning the porosity of the carbon materials and utilizing novel nonaqueous electrolytes to increase the operating voltage of the supercapacitor [10,13,24,25]. It has been reported that the capacitance of asphalt-derived carbon materials demonstrates 127 to 140 F/g for discharge currents of 0.5 to 10 A/g using aqueous KOH electrolytes [26]. This is a promising result for using low-cost starting materials such as asphalts for energy-storage applications. However, the processes used in that particular work are multistep and tedious. It is of our interest herein to use simple straightforward chemical processes to modify the porosity, defect density, conductivity, and architecture of asphalt-based materials for symmetric supercapacitor applications.

Two main issues with asphalt-derived carbons related to supercapacitor applications are pore size distribution and electrical conductivity. The carbonization and activation processes performed on asphalts typically resulted in microporosity that significantly increases the BET surface area. However, as the supercapacitor energy storage is usually performed in liquid electrolyte media, access to the micropores is often hindered, especially when using high discharge currents. Furthermore, high annealing temperature (>800 °C) is usually employed to improve the electrical conductivity. Typically, when the annealing temperature is too high, the micropores collapse, dropping the surface area significantly. Due to the target activation temperature of 400 °C, the main focus of this work was on the

effect of microporous structure and the effect of activating agents; in particular, $Ca(OH)_2$ and MgO as the main components of industrial waste. VR-KOH revealed the highest surface area with a predominantly microporous structure. Three-electrode setups with $KNO_3$ as an electrolyte were used to test the performance of the electrochemical super-capacitors, although due to the higher ionic conductivity of hydroxyl ($OH^-$) ions, the KOH electrolyte is expected to give a better capacitance than $KNO_3$. However, a neutral electrolyte such as $KNO_3$ is environmentally benign, which is in line with our objective of sustainability to utilize petrochemical waste. In addition, neutral electrolytes have been shown to expand the electrochemical window for carbon-based supercapacitors [27,28]. Specific capacitance (Csp) of the electrodes derived from VR-activated carbons was estimated using the expression shown below:

$$Csp = I/\{V \times (dv/dt) \times m\}$$

where $I$ is the CV curve area, dv/dt is the scan rate, $V$ is the potential range, and $m$ is the mass of the active carbon materials. Supercapacitors require an accessible surface area; therefore, the high specific capacitance of 91.91 F/g was achieved for VR-KOH. On the other hand, carbon black with a specific surface area of 266 $m^2$/g shows a specific capacitance of 48.21 F/g; see Figure 6 and Table 3. The rate performance data of different carbon electrodes in a 2-electrode cell configuration is shown in Figure 6c and was collected from CV performance at different scan rates. For all the samples, the trend shows a decreasing capacitance with increasing scan rates. The result is in agreement with Table 3, in which the performance of VR-$Ca(OH)_2$ is comparable to the carbon black reference sample. Furthermore, VR-KOH and VR-MgO, two samples with a large difference in microporous areas, show a comparable specific capacitance, especially at higher scan rates. The potential range showed a minimum non-faradaic current according to a 3-electrode configuration test for each sample, which is indicative of the cyclability of the activated VR electrodes. Thus, it is not expected for any significant decay in capacity due to the electrochemical side reaction of the electrode and electrolyte. Nevertheless, capacity loss due to the mechanical detachment of the carbon powders is possible. Systematic investigation with XPS and SEM-EDX of the electrodes before and after cycling will give a better picture of the long-term electrochemical stability of the activated carbons. However, this is not the focus of our manuscript, which mainly highlights activating petroleum waste and its potential application. As an indication of the short-term cycle stability of VR-KOH, we provided cyclic voltammetry data of the VR-KOH electrode in a symmetric 2-electrode configuration showing similar current values for cycles 1 and 15 (Figure 6d).

**Table 3.** Summary of specific capacitance analyzed from 2-electrode cell measurements.

| Sample | Device Capacitance * (F), $\times 10^{-3}$ | Electrode-Specific Capacitance (F), $\times 10^{-2}$ | Powder-Specific Capacitance (F/g) | Carbon-Specific Capacitance (F/g) | Carbon Content%, from XPS Data |
|---|---|---|---|---|---|
| VR-KOH | 6.00 | 2.40 | 91.91 | 108.13 | 85.00 |
| VR-$Ca(OH)_2$ | 5.10 | 2.04 | 17.06 | 39.67 | 43.00 |
| VR-MgO | 4.10 | 1.64 | 27.39 | 74.01 | 37.00 |
| Carbon black | 8.20 | 3.28 | 48.21 | 48.21 | 100.00 |

* Values were calculated from the slopes of the scan rate versus the average current plot. Repeated CV measurements show about a 4% of the standard deviation of current values at 100 mV/sec and 0.4 V.

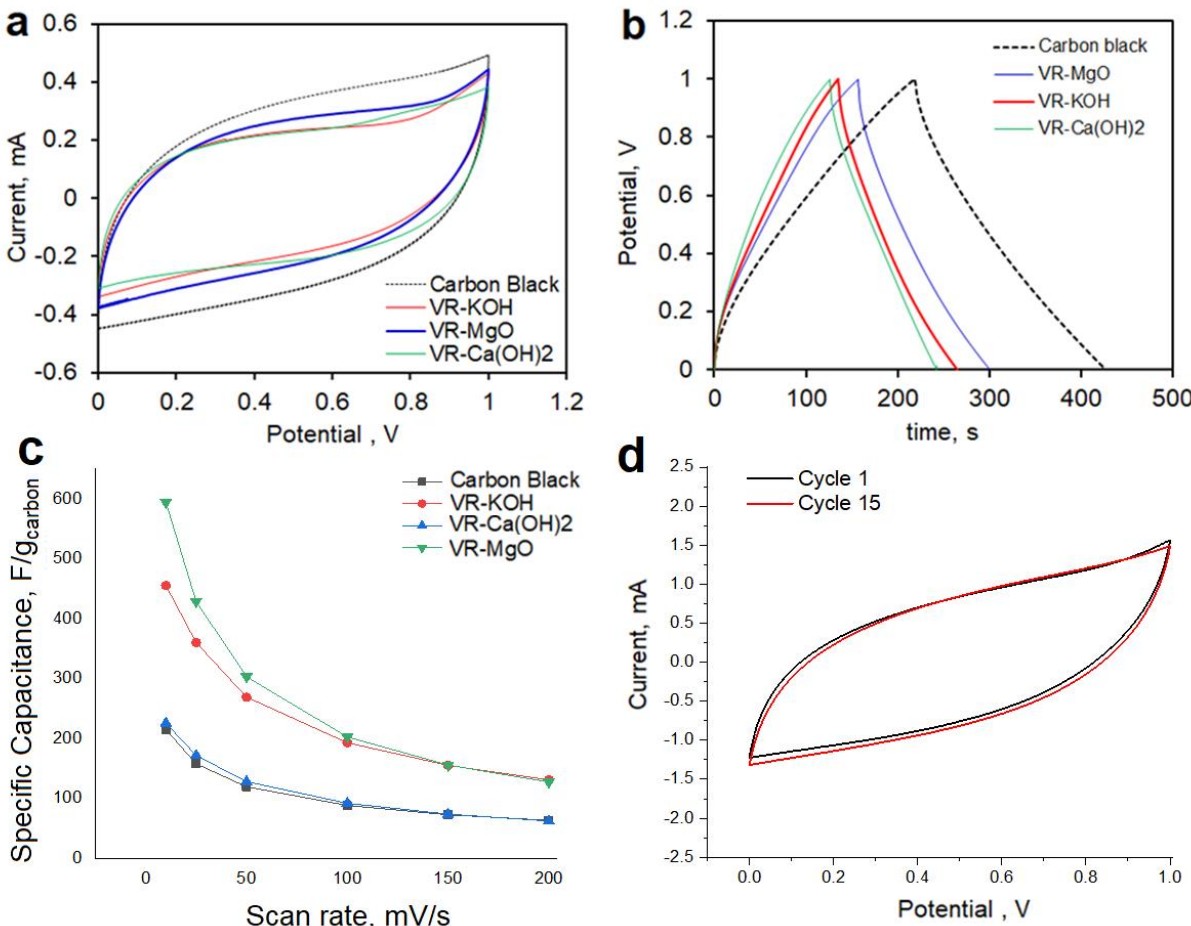

**Figure 6.** (**a**) Cyclic voltammetry of carbon black, VR-KOH, VR-Ca(OH)$_2$, and VR-MgO at 10 mV/s scan rate; (**b**) charge–discharge curves at a current density of 0.200 mA in 1 M KNO$_3$ electrolyte; (**c**) variation of specific capacitance with a scan rate of carbon electrode performed in 2-electrode cell configurations in 1 M KNO$_3$ electrolyte; and (**d**) cyclability of VR-KOH electrode in 1 M KNO$_3$ electrolyte.

The fact that both VR-Ca(OH)$_2$ and VR-MgO exhibit similar capacitance with carbon black, and that VR-KOH shows only about a 50% increase in capacitance indicates that the capacitance mainly comes from macropores. So, the micropores in KOH were not fully utilized. This could be due to the nature of the ink binder used for electrode preparation, the cell assembly, or it just being the intrinsic nature of the micropores that are too small for the solvated ions in the electrolyte. Further details of the importance of the electrode preparation are beyond the scope of the work and will be addressed in future reports.

## 4. Conclusions

In summary, vacuum residue directly acquired from the extraction facilities of the local industry was used to prepare activated carbons using various activating agents, such as KOH, Ca(OH)$_2$, and MgO. Characterizations of the final activated carbons derived from the vacuum residue resulted in the highest porosity for VR-KOH. Ca(OH)$_2$ and MgO as activating agents might require higher temperatures. However, the electrochemical supercapacitor performances of VR-Ca(OH)$_2$ and VR-MgO reveal being comparable with carbon black, which is typically made at >900 °C of the activation temperature. When the powder capacitance values are normalized with their corresponding BET surface area values, VR-KOH ranks the lowest, while VR-MgO and VR-Ca(OH)$_2$ exhibit higher numbers similar to the nonporous carbon black. This comparison highlights the discrepancies of surfaces in micropores for the capacitive current generation. This indicates that the micropores in VR-KOH are too narrow for the hydrated K$^+$ and NO$_3^-$ species. Future

work to optimize the activation step to shift the porosity distribution in VR-KOH to larger diameters is expected to enhance the specific capacity of the activated carbon sample.

**Author Contributions:** Conceptualization, A.R. and A.S.J.; methodology, A.R. and A.S.J.; validation, A.A. (Abdualilah Albaiz), M.A., A.A. (Abdullah Alzahrani), A.R. and A.S.J.; formal analysis, A.A. (Abdualilah Albaiz), M.A. and A.A. (Abdullah Alzahrani); investigation, A.A. (Abdualilah Albaiz), M.A. and A.A. (Abdullah Alzahrani); resources, A.R. and A.S.J.; data curation, A.A. (Abdualilah Albaiz), M.A., A.A. (Abdullah Alzahrani), H.A., A.R. and A.S.J.; writing—original draft preparation, A.A. (Abdualilah Albaiz), M.A., A.A. (Abdullah Alzahrani), A.R. and A.S.J.; writing—review and editing, A.R. and A.S.J.; visualization, A.R. and A.S.J.; supervision, A.R. and A.S.J.; project administration, A.R. and A.S.J.; funding acquisition, A.S.J. All authors have read and agreed to the published version of the manuscript.

**Funding:** King Fahd University of Petroleum and Minerals (KFUPM), the Deanship of Research Oversight, and Coordination funding project DF191019.

**Data Availability Statement:** The data is available upon request from the corresponding authors.

**Acknowledgments:** The authors thank the financial support of King Fahd University of Petroleum and Minerals (KFUPM), the Deanship of Research Oversight, and Coordination for funding this work through project DF191019.

**Conflicts of Interest:** The authors declare no conflict of interest.

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
