# Peer review of "Active Carbon-Based Electrode Materials from Petroleum Waste for Supercapacitors"

_carbon, 2022_

Round 1

Reviewer 1 Report

The authors were able to develop activated carbon electrodes successfully. The way the study and discussions were explained was rational and straightforward. But there are some questions and comments below. After these issues have been resolved, the paper can be published.

1.       What CDL stands for in line 47?

2.       Please check the whole manuscript for some typos and mistakes, such as line 198 “400 0C” and line 245 “more less”

3.       To have a better introduction, you can name some other materials that have been used as the electrode of supercapacitors recently (https://doi.org/10.3390/en15031162, https://doi.org/10.1002/adfm.202110267)  

4.       The indicated scale bars in Figure 2 are not clear enough. You can write them down in the caption.

5.       Have you done EDX analysis to track the residues on activated carbon samples?

Author Response

We would like to thank the Reviewer for the valuable comments and suggestions. We have accepted and made all changes to the manuscript. Changes made are given in yellow.

Comments from Reviewer:

  1. ‌What CDL stands for in line 47? Fixed
  2. Please check the whole manuscript for some typos and mistakes, such as line 198 “400 0C” and line 245 “more less” Fixed
  3. To have a better introduction, you can name some other materials that have been used as the electrode of supercapacitors recently (https://doi.org/10.3390/en15031162, https://doi.org/10.1002/adfm.202110267) Added as new references 
  4. The indicated scale bars in Figure 2 are not clear enough. You can write them down in the caption. Fixed
  5. Have you done EDX analysis to track the residues on activated carbon samples? Yes, we did EDX analysis and discussed it on page 5.

Reviewer 2 Report

This paper reports the fabrication of carbon electrode-based materials derived from petroleum waste by carbonization followed by activation with different bases. The prepared carbon materials have been characterized by the authors using various characterization techniques, including XPS, BET, SEM and TGA. Among the tested carbon samples, VR-KOH showed the highest specific capacitance of 108 F/g. Overall, the work shows the transformation of waste products, such as vacuum residue into valuable electrode materials for supercapacitors, which is interesting to the readers. Hence, the work is suitable for publication in this journal. However, some revisions are needed before this work can be accepted as detailed below:

1. How about the degree of graphitization of the different carbon samples? Please check by TGA and discuss the results.

2. In Figure 4, the peaks in the survey spectra of the carbon samples need to be labelled.

3. The Conclusion section is too brief and need more explanation as to why VR-KOH showed the best electrochemical performance for supercapacitors.

4. Is there any reason why KNO3 is used as the electrolyte than standard KOH?

5. The rate capability of the electrode needs to be tested and discussed, e.g., what are the specific capacitance values at different current density?

6. The cyclability of VR-KOH should be checked as well.

7. In Fig. 2, it is better if the original scale bars on the SEM images are deleted and the scale bars redrawn by the authors.

8. In the Experimental section, it is better if the synthesis procedure description is separated from the list of materials/chemicals under a separate heading.

9. In Materials section, please include the purity of the chemicals used.

10. In the Introduction, more recent and up-to-date references on the development of carbon-based electrodes for supercapacitors, such as Ceramics Int., 48, 9197-9204; Journal of Energy Storage, 50, 104224(2022) and Nano Convergence, 9, 10 (2022) can be included and cited.

11. The English grammar in this manuscript can be further improved.

Author Response

We would like to thank the Reviewer for the valuable comments and suggestions. We have accepted and made all changes to the manuscript. Changes made are given in yellow.

Comments from Reviewer:

  1. How about the degree of graphitization of the different carbon samples? Please check by TGA and discuss the results. The degree of graphitization of the sample can not be assessed by TGA measurements. This is because the quantity and type of alkali and alkali earth metal oxides in the activated samples are different. The oxidation kinetics of the carbon in the sample will be affected by these metal oxides. However, we performed Raman measurement as an indication of degree of graphitization after the low temperature activation step. The Raman spectra presented herein compares VR-KOH in Figure 5, as the sample with the highest carbon yield and most porous with graphite powder.

2. In Figure 4, the peaks in the survey spectra of the carbon samples need to be labelled. Fixed

3. The Conclusion section is too brief and need more explanation as to why VR-KOH showed the best electrochemical performance for supercapacitors. Fixed

4. Is there any reason why KNO3 is used as the electrolyte than standard KOH? We have addressed this on page 8 with additional references.

5. The rate capability of the electrode needs to be tested and discussed, e.g., what are the specific capacitance values at different current density? Added new date with discussion. in Figure 6.

6. The cyclability of VR-KOH should be checked as well.  Added new date with discussion. in Figure 6.

7. In Fig. 2, it is better if the original scale bars on the SEM images are deleted and the scale bars redrawn by the authors. Fixed

8. In the Experimental section, it is better if the synthesis procedure description is separated from the list of materials/chemicals under a separate heading. Fixed

9. In Materials section, please include the purity of the chemicals used. Fixed

10. In the Introduction, more recent and up-to-date references on the development of carbon-based electrodes for supercapacitors, such as Ceramics Int., 48, 9197-9204; Journal of Energy Storage, 50, 104224(2022) and Nano Convergence, 9, 10 (2022) can be included and cited. Added new references in the Introduction. Highlighted in yellow.

11. The English grammar in this manuscript can be further improved. Fixed

Reviewer 3 Report

Report on the manuscript: “Active carbon-based electrode materials from petroleum waste for supercapacitors”, A. Albaiz et al.

Ref. Carbon-2057572.

General comments:

The manuscript describes the preparation, characterization and performance as base materials for supercapacitors of carbonaceous materials fabricated from petroleum waste. This is a reasonable manuscript, although several aspects need clarification according to the following set of considerations.

General remarks:

I) The authors mention in the Experimental section “Vacuum residue from local industry” as the source of the petroleum waste. Obviously, major information is required and, logically, the organoleptic properties and composition –at least roughly- of the native material should be provided.

II) The authors focus their discussion exclusively on hydrocarbon materials. However, sulfur compounds should, in principle, also be present. This could be a factor to be accounted for.

III) Following the same line of reasoning, the carbon content in Table 2 is calculated from XPS data probably accounting C-C and C-O linkages. This procedure is problematic and avoids the possible contribution of other elements. Logically, SEM/EDX data should be more appropriate for the semiquantitative purposes and would offer information on the aforementioned sufur compounds.

IV) As is unfortunately frequent in recent literature, the authors tend to express all numerical quantities with an unrealistic number of significant figures if replicate experiments are carried out. For instance, writing a micropore area of 1164 units (Table 1) means that this quantity is known with a relative uncertainty of only 0.09 % (!!). All quantities in Tables 1 and 2 should be accompanied by their corresponding standard deviations, all quantities being written with the pertinent number of significant figures according to the usual IUPAC recommendations.

Author Response

We would like to thank the Reviewer for the valuable comments and suggestions. We have accepted and made all changes to the manuscript. Changes made are given in yellow.

Comments from Reviewer:

I) The authors mention in the Experimental section “Vacuum residue from local industry” as the source of the petroleum waste. Obviously, major information is required and, logically, the organoleptic properties and composition –at least roughly- of the native material should be provided.

II) The authors focus their discussion exclusively on hydrocarbon materials. However, sulfur compounds should, in principle, also be present. This could be a factor to be accounted for.

 We have addressed this by giving a quantitative analysis of the XPS analysis in the new table 2 and discussed the EDX analysis in the text given in yellow color in page 5.

III) Following the same line of reasoning, the carbon content in Table 2 is calculated from XPS data probably accounting C-C and C-O linkages. This procedure is problematic and avoids the possible contribution of other elements. Logically, SEM/EDX data should be more appropriate for the semiquantitative purposes and would offer information on the aforementioned sufur compounds.

We thank Revewer for the valuable comment. Similar to the previous comment by the reviewer.  We have addressed this by giving a quantitative analysis of the XPS analysis in the new table 2 and discussed the EDX analysis in the text given in yellow color in page 5. We performed EDX measurements at 20 kV to address the EDX-average value of the samples’ sulfur content. The measurements on all of the activated samples show low content of sulfur. VR-MgO sample showed the highest sulfur content around 1.0 wt%. While both VR-KOH and VR-Ca(OH)2 samples show less than 0.1 wt%. EDX measurements were performed with a minimum of 5 spots for every sample. Thus, due to the low amount of sulfur in the activated sample compared to oxygen, we believe sulfur contribution to the overall capacitance is insignificant.

IV) As is unfortunately frequent in recent literature, the authors tend to express all numerical quantities with an unrealistic number of significant figures if replicate experiments are carried out. For instance, writing a micropore area of 1164 units (Table 1) means that this quantity is known with a relative uncertainty of only 0.09 % (!!). All quantities in Tables 1 and 2 should be accompanied by their corresponding standard deviations, all quantities being written with the pertinent number of significant figures according to the usual IUPAC recommendations.

From several N2 isotherms of a representative sample, we found that the of BET surface area values has standard deviation of 14%. This may require more statistics to better represent high and low surface area range. However, we do not have such elaborate statistics. Nevertheless, the variation of surface areas within the four samples tested herein exceed the 14% standard variation. Hence, the claims we made in the manuscript regarding surface areas should be acceptable. We have changed the values in Table 1 accordingly.

Round 2

Reviewer 2 Report

The authors have addressed my previous comments thoroughly and conducted additional experiments needed to improve the quality of this paper. Hence, I am happy to accept the paper in the present form.

Author Response

Please see attached cover letter
